
# 1  An atmospheric approach to the flood disaster in the Western
# 2  Black Sea region (Turkey) on 10-12 August 2021

**Onur Halis[1], Barbaros Gönençgil[2], Zahide Acar[3]**
[1]Geography Department, Istanbul University, Istanbul, 34000, Turkey.
[2]Geography Department, Istanbul University, Istanbul, 34000, Turkey.
[3]Geography Department, Çanakkale Onsekiz Mart University, Canakkale, 17000, Turkey.
Correspondence to: Onur Halis (onurhalis@istanbul.edu.tr)
**Abstract**. Precipitation is one of the important climate elements that directly affect disasters with its high spatial and
temporal variability. In addition to the changes in precipitation amounts between years, precipitation intensity and
duration are also necessary for precipitation climatology. The Black Sea coast is one of the regions that attract attention
in terms of precipitation amounts, intensity and duration in the world. Countries bordering the Black Sea have been
struggling with flash floods, which have arisen due to excessive rainfall in recent years. The flood has had the most
loss of life and devastating effect in recent times, especially in the summer season, which occurred after heavy rains
on August 10-12, 2021. In this disaster, 82 people lost their lives due to the flood. In addition, the flood damaged
energy, transportation, access to drinking water, communication, and shelter areas. From 10 to August 12, total
accumulated rainfall amounts of 452.5, 389.2, 386.2 and 281.1 mm were measured in the Bozkurt Mamatlar Village,
Devrekani Kuzköy, Küre and Abana settlements of Kastamonu city, respectively. Additionally, precipitation was
recorded 235.9 mm in Ulus Ceyüpler Village of Bartın, 317.9 mm in Ayancık and 225.8 mm in Türkeli of Sinop. In
this study, the atmospheric conditions of the flood disaster were examined. The Basra low pressure settled over the
eastern Black Sea during this period. On August 8-9, a trough developed above Anatolia at the level of 500 hPa. Due
to the low pressure remaining on the region, the system coming to the Black Sea Basin from the north turned towards
the east of the Black Sea as of August 8. Especially at this date, while high-pressure conditions continued in most
western parts of the Black Sea, the pressure falling from the Sevastopol-Kastamonu line to the east attracted the system
coming from the north. On August 10 2021, the surface temperature of the Black Sea, especially the eastern part,
reached 28.3 °C (3.5 °C higher than average). The temperature difference between sea surface temperatures and 850
hPa reached 13 °C degrees. These conditions resulted in a strong low-level moisture convergence development that
continued for three days in the Black Sea Basin. Thus, the developing convective cell gradually became more vigorous,
expanded its area of influence in the interior of the land due to the cyclone movement as of the morning of August 10
and caused heavy rains with the effect of the topography.



## 1 Introduction

Precipitation is one of the most variable elements of the climate and is controlled by factors such as geographical location, general atmospheric circulation and local conditions (Erinç, 1996; Erol, 2011). In addition to these, it can be said that the trends in extreme precipitation have a more sensitive response to climate change than other climate elements. As a result of this situation, extreme climate events attract more attention (Katz & Brown, 1992). For example, Bocheva et al. (2009), in their study covering the whole of Bulgaria, compared the total precipitation and heavy precipitation in the two periods, 1961-1990 and 1991-2005 and found that a significant amount of daily heavy precipitation was experienced in the second period. Especially in Bulgaria, due to heavy rains in 2005, 25 people died their lives and caused more than 500 million Euros of damage.

Excessive rainfall and the mechanisms that cause this rainfall lead to disasters such as landslides and floods, causing socio-economic problems and, most importantly loss of life. In particular, the rugged topography of the eastern part of the Black Sea Basin, the suddenly rising mountains, and the interaction between the sea and the land has led to increased rainfall and devastating floods in the countries bordering the Black Sea in recent years. For example, the flood on 6-7 July 2012 in Krymsk, Novorossiysk and Glendzhik in the Krasnodar Region of Russia caused approximately 170 deaths and USD 625 million in damage (Alexeevsky et al., 2016; Kapochkina et al., 2015). On August 22, 2012, due to heavy rains in the northeast of the Black Sea, a flood occurred in the Tuapse region of Russia (Alexeevsky et al., 2016). On June 20, 1988, 179 mm of precipitation in 4 hours and 50 minutes was measured in Novorossiysk in the Krasnodar Region of Russia. It was stated that the water hose was also influential in this precipitation (Alexeevsky et al., 2016; Kapochkina et al., 2015). In flood triggered by the development of the convective cell in the Mediterranean cyclone, the total amount of precipitation on 24–25 October varied from 253 to 362 mm in the Tuapse River basin and 208.9 mm in the Shakhe River basin to 53.2 mm near the Sochi River mouth and 83.6-112 mm in the Mzymta River basin (Korshenko et al., 2020). One of the countries affected by floods due to heavy rains in this region is Turkey. Yüksek et al. (2013) examined 51 significant floods in the Eastern Black Sea Basin of Turkey between 1955 and 2005 years, and they stated that 258 deaths occurred as a result of these floods; the damage was close to the US $500,000,000. On August 24, 2015, located on Turkey's Eastern Black Sea coast, 136 mm in Hopa, 64 mm in Arhavi and 109 mm in precipitation in Borcka were measured. Eleven people lost their lives in the flood, and the economic loss reached 1 million dollars (Baltaci, 2017). One of the most significant reasons for this excessive rainfall is a positive anomaly of the sea surface temperature (SST) over the Black Sea (Baltaci, 2017; Doğan et al., 2019). Extreme rains, the effect of which gradually increases in the summer seasons, have recently




caused more than one flood. In this study, we tried to examine the atmospheric conditions of the hourly and daily
precipitation totals that occurred on 10-12 August 2021 in the provinces of Sinop, Kastamonu and Bartın on the Black
Sea coast. Within the scope of the study, answers to the following questions were sought:
• If there is an increase in the number and severity of atmospheric originated floods on the Black Sea coast,
what factors cause this?
• How is the temperature gradient of the upper and lower atmospheres with the surface temperatures?
• What are the trends in changes in sea surface temperatures?
• What is the role of the gradually warming sea surface temperature in the Black Sea in heavy rains?
• What is the role of topography in the variation of precipitation intensity?
The answers to the above questions are essential in understanding and predicting the atmospheric mechanisms that
cause excessive precipitation in the Black Sea coast of Turkey.
**2  Study Area**
The provinces of Sinop, Kastamonu and Bartın, located in the Western Black Sea region of Turkey and stretch along
the Black Sea coast for 420 km, constitute the study area (Fig. 1). In these provinces, where approximately 800
thousand people live, there are many settlements in the river valleys (Turkish Statistical Institute, 2021). The fact that
the topographic relief does not show a homogeneous feature, suddenly changing altitude and slope conditions cause
uneven distribution of precipitation. Especially within 15 km of the coast, the altitude rises above 1000 meters. The
Küre Mountains, stretching between Bartın and Sinop and reaching a height of 2019 meters (Yaralıgöz Peak), are the
first mountain ranges in which the air masses coming from the Black Sea are forced to rise in a short distance. The
second mountain range to the south of these mountains, separated by plateaus and depressions, is the Ilgaz Mountains
at 2587 meters (Büyükhacat Hill). Most of the land within the study area, including these depressions and plateaus, is
above 1000 meters in elevation. When examined from a hydrological point of view, many short rivers take their source
from the Küre Mountains and flow into the Black Sea. The narrow and deep valleys opened by these rivers form
critical units in which the air masses are forced to rise by canalization and positively affect precipitation. This situation
played an essential role in the flood in the Bartın, Ezine and Ayancık Streams after extreme rains in the study area on
10-12 August.

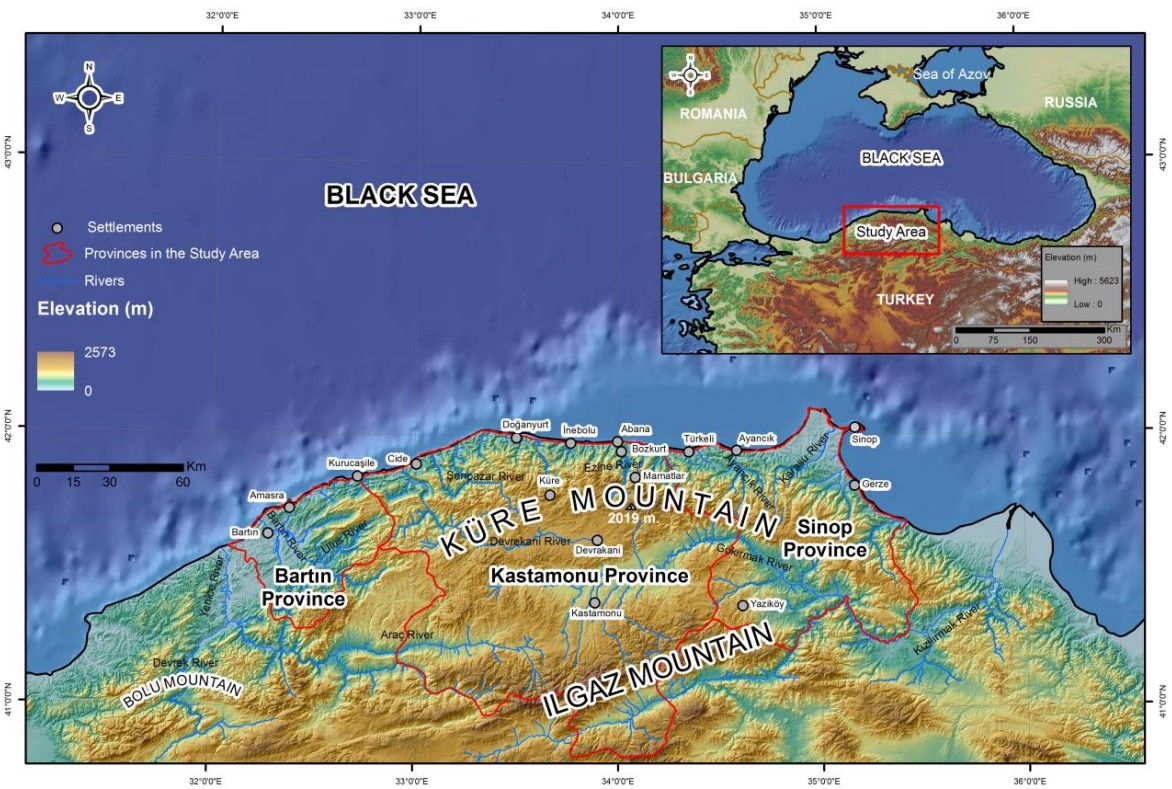

**Figure 1.** Provincial borders constituting the study area, Western Black Sea Region.

### 3 Data and Method

In order to evaluate the extreme conditions in the study area, hourly and daily precipitation data of a total of 56 stations located in Sinop, Kastamonu and Bartın were used (Table 1, fig. 11, 12 and 13.). The ground charts, 850, 700 and 500 hPa synoptic maps used in the study were obtained from MGM (Turkish State Meteorological Service). In this study, long-term and daily NOAA High-Resolution Blended Analysis data from 1971-2000, 1991-2020 and 10-17 August 2021 were used to examine the role of sea surface temperatures in extreme precipitation (Reynolds et al., 2007). On 10-11-12 August 2021, Natural Color RGB, Day Microphysics RGB, Convection RGB data from Meteosat Second Generation were used to determine the spatial distribution of convective cells and to examine atmospheric conditions. In addition, SYNOP observations and NOAA's daily temperature data and 700 hPa u and v direction vector wind data were also used with these data.


**Table 1:** Description of the 56 meteorological stations included in the study and daily precipitation data. The stations marked with a star were evaluated in terms of climatology.

| No, Station Code and Name | | Longitude (°E) | Latitude (°N) | Altitude (m) | 10 Aug 2021 precipitation | 11 Aug 2021 precipitation | 12 Aug 2021 precipitation |
|---|---|---|---|---|---|---|---|
| 1 | 19031 - Ağlı | 33.55440 | 41.68440 | 1190 | 54.4 | 74.5 | 2.6 |
| 2 | 18215 - Araç | 33.33060 | 41.24670 | 682 | 7.8 | 95.1 | 2.2 |
| 3 | 18705 - Araç Orman S. | 33.35166 | 41.20388 | 1240 | 9.8 | 97.9 | 2.3 |
| 4 | 18514 - Azdavay | 33.28360 | 41.63610 | 822 | 46.5 | 108.8 | 2.2 |
| 5 | 19032 - Mamatlar | 34.12750 | 41.79470 | 1523 | 121.7 | 297.6 | 33.2 |
| 6 | 17604 – Cide* | 32.94760 | 41.88220 | 42 | 2 | 40.4 | 5.4 |
| 7 | 17625 – Çatalzeytin* | 34.21800 | 41.95380 | 9 | 73.2 | 6 | 0 |
| 8 | 18515 – Daday | 33.43923 | 41.50068 | 1009 | 12.4 | 48.3 | 0.1 |
| 9 | 19030 - kuzköy | 34.04420 | 41.77280 | 1436 | 93.4 | 262.7 | 33.1 |
| 10 | 17618 – Devrekani* | 33.83450 | 41.59960 | 1099 | 50 | 92 | 0.6 |
| 11 | 18516 – Doğanyurt | 33.45970 | 42.00670 | 7 | 25.5 | 1.3 | 6 |
| 12 | 18517 – İhsangazi | 33.54060 | 41.21860 | 850 | 15.2 | 36.2 | 2 |
| 13 | 18704 – İncigez | 33.48110 | 41.25860 | 1231 | 24.1 | 54.3 | 10 |
| 14 | 17024 – İnebolu* | 33.76360 | 41.97890 | 49 | 12.4 | 34.8 | 24.6 |
| 15 | 18911 – İnebolu kar | 33.71890 | 41.87470 | 960 | 115.9 | 137.4 | 21.7 |
| 16 | 18914 – Yolüstü | 33.73366 | 41.91193 | 729 | 70 | 128.6 | 22.4 |
| 17 | 17074 – Kastamonu* | 33.77560 | 41.37100 | 800 | 19.9 | 49 | 7.2 |
| 18 | 18706 - Başören | 34.11720 | 41.29440 | 961 | 45.7 | 50.2 | 6.3 |
| 19 | 17606 – Bozkurt* | 34.00370 | 41.95970 | 149 | 119.1 | 36.5 | 7.5 |
| 20 | 18997 - Mescit | 33.89898 | 41.24982 | 1149 | 37.7 | 23 | 12.5 |
| 21 | 18513 – Pınarbaşı | 33.12440 | 41.61780 | 644 | 56.9 | 92 | 2.2 |
| 22 | 18518 – Küre | 33.71110 | 41.81060 | 1013 | 173.1 | 201.1 | 12 |
| 23 | 18520 – Seydiler | 33.71860 | 41.62440 | 1052 | 49.9 | 55.9 | 0.4 |
| 24 | 18521 – Şenpazar | 33.23000 | 41.81080 | 395 | 23.5 | 50.7 | 12.3 |
| 25 | 18522 – Taşköprü | 34.22250 | 41.49420 | 632 | 41.3 | 48.6 | 3.3 |
| 26 | 17650 – Tosya* | 41.01320 | 33.23000 | 860 | 23.6 | 14.6 | 0 |
| 27 | 19221 – Abana | 34.00006 | 41.97741 | 9 | 122.6 | 141.7 | 16.8 |
| 28 | 19222 – Hanönu | 34.50010 | 41.62828 | 430 | 36.8 | 74.1 | 10.1 |
| 29 | 18519- Pınarbaşı Kar | 33.07890 | 41.55660 | 1019 | 15.9 | 72 | 7.4 |
| 30 | 17602 – Amasra | 32.38270 | 41.75260 | 50 | 0 | 0 | 38.2 |
| 31 | 17020 – Bartın* | 32.35690 | 41.62480 | 36 | 0 | 0 | 34.6 |
| 32 | 17721 - Bartın/Arıt | 32.61560 | 41.68720 | 356 | 25.4 | 6.8 | 52.3 |
| 33 | 18692 - Hasankadı | 32.47500 | 41.35140 | 1264 | 12.1 | 40.4 | 18.2 |
| 34 | 19007 - Kozcağız | 32.34250 | 41.49440 | 51 | 0 | 24.9 | 34 |
| 35 | 18245 - Kurucaşile | 32.73603 | 41.80513 | 126 | 3 | 1.4 | 30.1 |
| 36 | 17615 – Ulus* | 32.63700 | 41.58190 | 182 | 57.6 | 71.6 | 21.6 |
| 37 | 19008 - Ulus/Ceyüpler | 32.54420 | 41.47080 | 630 | 95.7 | 96.4 | 43.8 |
| 38 | 19009 - Ulus/Çubukeli | 32.84940 | 41.69060 | 520 | 14.6 | 122.1 | 23.2 |
| 39 | 19207 – Kurucaşile | 32.73603 | 41.80513 | 184 | 8.5 | 5.1 | 38.8 |
| 40 | 19039 – Ayancık | 34.58110 | 41.92440 | 36 | 158.5 | 129.3 | 30.1 |
| 41 | 18546 - Akören | 34.77140 | 41.83220 | 625 | 62.7 | 138 | 13.1 |


| 42 | 19040 - Çangal | 34.65220 | 41.72060 | 1067 | 64.4 | 145.9 | 11.2 |
| 43 | 17620 – Boyabat* | 34.78530 | 41.46300 | 347 | 20.2 | 70.6 | 2 |
| 44 | 18711- Dranaz | 34.88530 | 41.62030 | 1109 | 32.1 | 71.5 | 6 |
| 45 | 19043 - Kavacık | 34.58860 | 41.37830 | 1130 | 27.6 | 66.5 | 0.5 |
| 46 | 18547 – Dikmen | 35.27250 | 41.66360 | 383 | 45.1 | 17.6 | 22.7 |
| 47 | 19042 - Dikmen | 35.25390 | 41.53030 | 1003 | 42.5 | 17 | 3.5 |
| 48 | 18548 – Durağan | 35.05060 | 41.43440 | 286 | 19.7 | 25.1 | 0 |
| 49 | 19041 - Durağan/Tekir | 35.29810 | 41.34640 | 997 | 21.2 | 11.8 | 26.6 |
| 50 | 18136 – Erfelek | 34.89470 | 41.87940 | 190 | 78.7 | 51.9 | 38.3 |
| 51 | 18549 – Gerze | 35.17360 | 41.81190 | 87 | 65.2 | 10.3 | 61.1 |
| 52 | 18550 – Saraydüzü | 34.84970 | 41.32850 | 422 | 4.8 | 47.9 | 0 |
| 53 | 17026 – Sinop* | 35.15450 | 42.02990 | 29 | 111 | 10.6 | 32.8 |
| 54 | 18551 – Türkeli | 34.33420 | 41.94110 | 123 | 82.1 | 143.7 | 0 |
| 55 | 19227- Sinop/Sarıkum | 34.92277 | 42.02472 | 21 | 86 | 32.8 | 19.7 |
| 56 | 19229 - Sinop/Sazlı | 34.87399 | 41.72357 | 1030 | 44.6 | 103.4 | 7 |

## 4 Findings

### 4.1 Synoptic Status

In terms of representing the general synoptic conditions of the study area, Karaca et al. (2000), as a result of the analysis of 15-year daily surface and 500 hPa synoptic maps, revealed that there are four main cyclone routes affecting Turkey (Fig. 2). The fact that especially the first two of these ways are effective on the Black Sea is vital in terms of the general synoptic conditions of the study area. The first of these originate in the southwest of Russia and passes over the Black Sea. The second originates from the Balkans and affects the Marmara and Black Sea Regions. These systems create intense precipitations throughout northern Turkey, effective in summer. Alexeevsky et al. (2016) also stated that the abnormal position in the northern location of the Azores anticyclone in recent years, with the movement of the anticyclone 1000-1500 km more north than usual, strong cyclogenesis from the Iberian Peninsula to the Balkans on the Mediterranean has become active. Thus the cyclones move from west to east along the Black Sea. In this study, meteorological conditions that caused flash floods in the provinces of Bartın, Kastamonu and Sinop on August 11 2021, were examined. In order to better understand the synoptic conditions of 10-12 August 2021, We started to evaluate the synoptic conditions of the day before the onset of heavy rains. When the synoptic situation at ground level is examined on August 9, it is seen that the Basra low pressure extends to the Eastern Black Sea Region (Fig. 3a). This situation created a low pressure over the entire Eastern Mediterranean, including Turkey. In the upper atmosphere, a ridge in the western Mediterranean and a trough developed over Turkey (Fig. 3b). In the 500 hPa meteorological maps, the winds developing following this trough formed northerly currents over the Western Black Sea and southerly currents over the Eastern Black Sea (Fig 4). In this case, the cold air entered from the northwest of the Black Sea. On August 10, the high-pressure area moved eastward over Europe and was located in the Northwest of the Black Sea (Fig. 3c). The blocking presence of the warm anticyclone over eastern Europe generated northerly


winds (Fig. 4). The distinguishing feature of this synoptic process is that it stopped the cyclone over the Black Sea.
As a result, the cyclone gained a stationary feature on August 10-12 (Fig. 3c, 3d and 4). The exact mechanism caused
floods on July 6-7, 2012, in Krymsk, Novorossiysk, and Glendzhik in the Krasnodar Territory (Alexeevsky et al.,
2016). Such abnormal pressure gradient is known to cause prolonged storms and trigger precipitation over very large
areas. (Fink et al., 2009). This situation increased the pressure gradient of the surrounding areas with the low pressure
developing on the middle parts of the Black Sea (the line between Crimea-Kastamonu), especially since August 10.
Thus, the Black Sea gained a vital convergence feature. The activation of the fronts on 09-12 August 2021, the
intensification of cyclogenesis on the Black Sea until about 40° latitude, and the presence of active front formation
are also noteworthy (Fig. 3a, 3c and 3d). This mechanism was reported on August 22, 2012, as the synoptic mechanism
behind the flood disaster in the Tuapse region (Alexeevsky et al., 2016). In addition to all these, the temperature
difference between sea surface temperatures and 850 hPa is around 13 °C (Fig. 3e). At 500 hPa, the temperature
gradient increased considerably (Fig. 3f). When the relatively cold northerly winds entered the Black Sea, they warmed
up from the bottom and the moisture content increased, resulting in an instability condition. In addition to these
synoptic processes, physiographic features (elevation, aspect, morphology) also played an important role in
precipitation variability and values at stations.

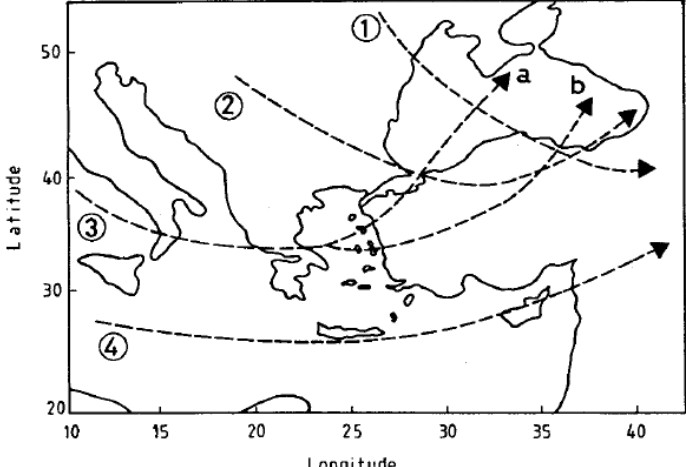


**Figure 2.** The paths of atmospheric cyclones over Turkey from Karaca et al. (2000).

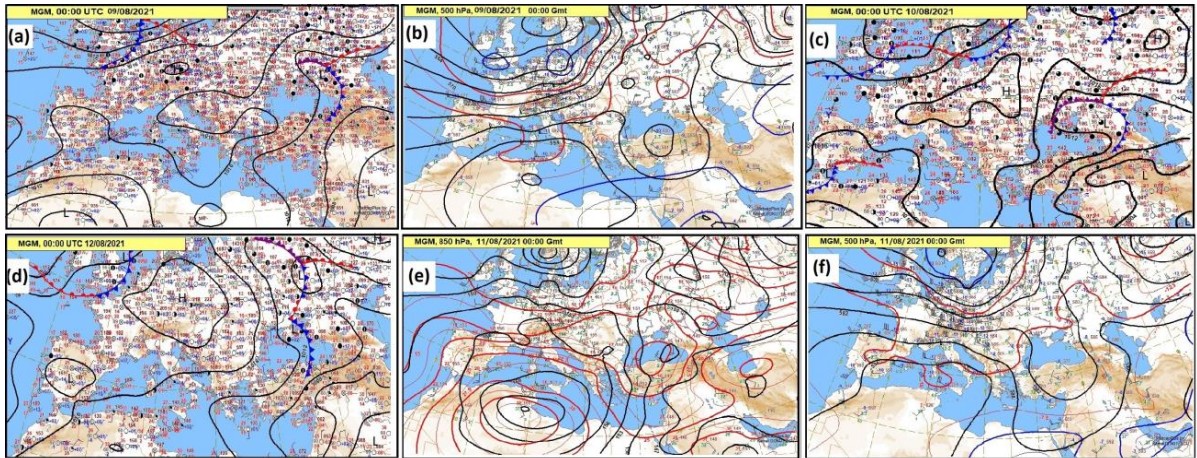

**Figure 3.** (**a**) Surface synoptic chart for August 9, 2021. (**b**) Geopotential heights at 500-hPa chart for August 9, 2021. (**c**) Surface synoptic chart for August 10, 2021. (**d**) Surface synoptic chart for August 12 2021. (**e**) 850 hPa synoptic chart for August 11, 2021, and (**f**) Geopotential heights at 500-hPa chart for August 11, 2021. Source: Turkish State Meteorological Service (www.mgm.gov.tr).

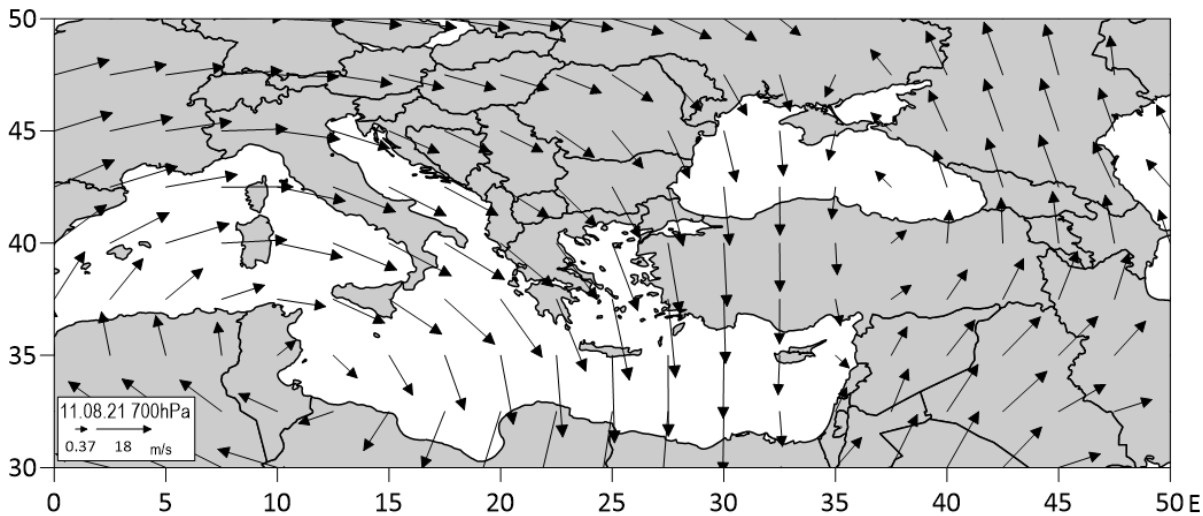

**Figure 4.** 11 August 2021, 700 hPa wind direction and speed (700 hPa u and v direction vector wind data.).





## 4.2. Satellite Data

On 10-11-12 August 2021, Natural Color RGB, Day Microphysics RGB, Convection RGB data from Meteosat Second Generation were used to determine the spatial distribution of convective cells and to examine atmospheric conditions. Natural Color RGB uses three solar channel spacings, red (NIR1.6), green (VIS0.8) and blue (VIS0.6). Water clouds with tiny droplets have a considerable reflection in all three channels and therefore appear whitish. Clouds with dense ice particles have a cyan colour and correspond to high clouds (EUMETSAT, 2021) (Figure 5a, 5b, 5c). VIS0.8 IR 3.9 IR 10.8 bands are used in Day Microphysics RGB images. In this way, convection, fog and low-level clouds can be easily observed. This is due to the clouds rising to the troposphere border, especially when the lower layers of the air mass are hot and strong vertical air movement develops. Cold and thick convective clouds with large ice particles on top, e.g. Cb tops, appear red (Figure 5d, 5e, 5f). This is important for detecting rising movements and more severe weather conditions. Another unique situation is that convective clouds are predominantly active when these images are examined in the study area. The Convection RGB combines the brightness temperature difference (BTD) between the WV6.2 and WV7.3 channels (on red), the BTD between the IR3.9 and IR10.8 channels (on green) and the reflectance difference between the NIR1.6 and the VIS0.6 channels (on blue) (EUMETSAT, 2021). Unlike Day Microphysics RGB, convection zones can be observed more easily in Day Convective Storms RGB. Severe convective storms appear bright yellow in this colour scheme (Figure 5g, 5h, 5i). High-level thick ice clouds and convective CB clouds containing large ice crystals appear red, and convective clouds containing tiny ice crystals appear yellow. When the Day Convective Storms RGB images of the Western Black Sea Region are examined, it can be said that convective CB clouds contain tiny ice crystals and strong ascending movements are dominant. (Fig. 5g, 5h and 5i). On August 10-11-12, it is seen that it develops in the north direction and moves southward and exhibits a cyclonic rotation. All these observations showed that precipitation concentrates on the sea and develops convectively. On the other hand, Doğan et al. (2019), according to the simulation results, it is seen that vertical cloud formation and updraft movement disappear as the sea surface temperature decreases. In addition, another result of this model is that it has been concluded that extreme precipitation will lose strength when SST is reduced by 2 °C.

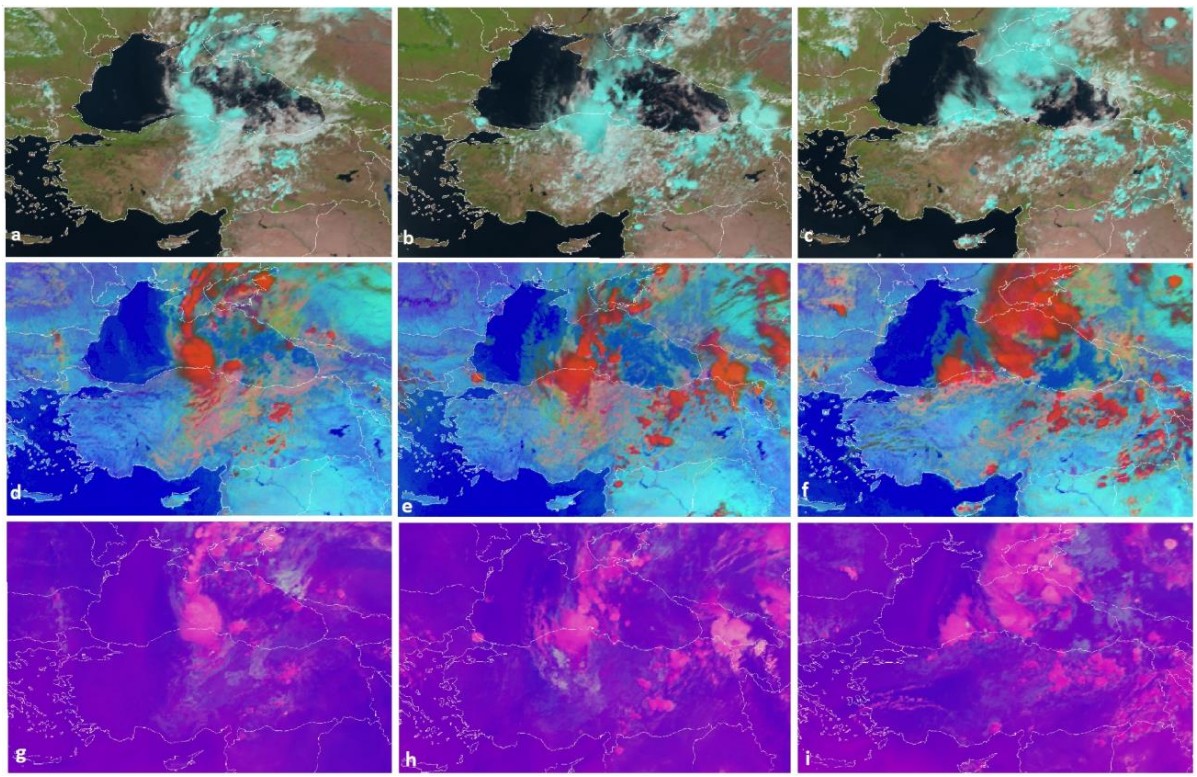

**Figure 5.** (a, b, c) Satellite images on 10-11-12 August 2021, 10:00 UTC, respectively. Natural Colour RGB product from SEVIRE MSG (Meteosat Second Generation). This satellite product reflects a cyan colour in ice-laden clouds. (d, e, f) Satellite images on 10-11-12 August 2021, 10:00 UTC, respectively. Cold and thick clouds with tops composed of large ice particles appear red. (g, h, i) Satellite images on 10-11-12 August 2021, 10:00 UTC, respectively, indicate severe convective storms. Satellite images on 10-11-12 August 2021, 12:00 UTC, respectively. Source: EUMETSAT (https://www.eumetsat.int/).

### 4.3. Sea Surface Temperatures

In the Black Sea Basin, some factors play a role in the fact that the average precipitation values on the northern coasts of Turkey do not decrease during the late summer and autumn transitions, especially in the extreme precipitation during these times. These factors should be related to the fact that the surrounding land areas are cooler than the sea, the absorption of solar radiation in the seas is high during these seasons and the landforms. Many studies have proved the effectiveness of seas or lakes on precipitation. For example, Pastor et al. (2001) investigated the relationship between torrential precipitation on the Mediterranean coast of Spain and sea surface temperatures (SST). They determined that SST was a critical factor in developing torrential precipitation in the Western Mediterranean Basin.



In the simulations made in Turkey, it is stated that the warm seas increase the precipitation in the Anatolian Peninsula,
especially the Black Sea, shows a significant difference in summer when compared to other seas, and the precipitation
tends to increase throughout the year with the increase in SST (Bozkurt & Sen, 2011). In our study, where the
temperature difference between the sea surface and the upper atmosphere is excellent, many studies emphasize the
effect of sea or lakes on precipitation where these temperature differences are emphasized (Suriano & Leathers, 2017).
West et al. (2019) pointed out that cloudiness and more precipitation occur over the Sea of Japan when the temperature
difference between the sea surface and the level of 850 hPa is 13 °C or more. Especially this mechanism is effective
in the west of Japan. Especially in cases where the temperature difference between the surface temperature and 850
hPa is 13 °C and above, the lake effect has been accepted as the threshold value for the development of precipitation
(Holroyd III, 1971; Kunkel et al., 2002; Niziol, 1987). According to Dai et al. (2018) simulated the lake effect in Nam
Co Lake based on data from four meteorological stations from October 2005 to December 2015 on the Tibetan Plateau.
They showed that it may have caused an increase of up to 60% in precipitation on the lake and the wind direction (east
of the lake) in October. In addition, this study stated that the lake surface was 15.5 °C warmer than the air at 550 hPa
in October. This temperature difference increased the instability conditions and caused convections. Wen et al. (2015)
examined the effects of Ngoring Lake and Gyaring Lake, which form the source of the Yellow River on the Tibetan
Plateau, local temperature and precipitation. It is stated that lakes contribute 49% to the annual precipitation on and
around the lake. This contribution is relatively high in the summer months and reaches 72%, especially from July to
October. In addition to lakes on the Tibetan Plateau, where similar conditions are experienced, lakes in Boreal regions
are also known to increase precipitation in summer (Eerola et al., 2010; Samuelsson et al., 2010). The above references
show that although the sea and lake effect is evident in different regions worldwide, the impact time and gradient are
significantly different.
As in the results of this study, the time interval where the lakes contribute the most to the precipitation around them
is night and morning hours when the water is warmer than the land (Fig. 10). In addition to all these, depending on
the NOAA data of August 10, 2021, the temperature values of the land areas in the Black Sea basin's surroundings,
and the daily minimum temperature values in the coastal areas of Romania, Ukraine, and Russia, Turkey were around
21 °C. On the other hand, it decreased to 15 °C in the inner parts where the altitude increased. According to MGM
data, this situation dynamically created a breeze of 3 m/s from land to sea due to the increase in the temperature
difference between the sea and the land during the night and morning hours. This triggered a low level of horizontal
convergence and allowed the air to rise. Thus, the divergence at the upper level contributes to moving the low-level
horizontal flow and humidity to the higher level. Thermally, the seas release more latent heat and contribute to
evaporation in developing convection for precipitation. This causes instability in the atmosphere.



Between 10-17 August 1971-2000, the maximum temperatures were 25 °C, while the minimum temperatures were
located northwest of the Black Sea and the Sea of Azov. It is noteworthy that the maximum temperatures between
1991 and 2020 exceeded 26 °C. In the sea surface temperatures of 10-17 August 2021, a temperature above 28 °C in
the east of the Black Sea is remarkable (fig. 6). There was a temperature anomaly of 3.5 °C in areas where convective
cells developed on days with extreme precipitation. As a source of evaporation, the warm sea in the north of the study
area has an essential effect on precipitation. In particular, it exceeded 27 °C in the central part of the Black Sea and
28 °C in the eastern part of August 10, 2021, SST (Fig. 6).
On the other hand, as seen in Fig. 6, the temperatures of the sections mentioned on August 13 decreased, and the hot
sections narrowed. This date is essential because it is the date when extreme precipitation ends. Especially in the south
of the Sea of Azov, where the convective cell developed, temperatures dropped to 25 °C on August 14 and 23 °C on
August 17. As a result of the evaporation of the water, the temperature was transferred from the sea surface to the
atmosphere, and the warm air was raised by convection. This situation continued between 10 and 13 August. The
south of the Azov Sea, about 27 °C before the start of precipitation, dropped to 23 °C on August 17, creating a
temperature difference of 5 degrees. As a result of such excessive precipitation, temperature differences of 3-5 °C
occur on the sea surfaces (Pastor et al., 2001). Another well-known result is the deepening of the low-pressure centre
as sea surface temperatures increase in the Eastern Black Sea Region (Doğan et al., 2019). This event is vital in
explaining both the role of the Black Sea and the source of excessive precipitation.

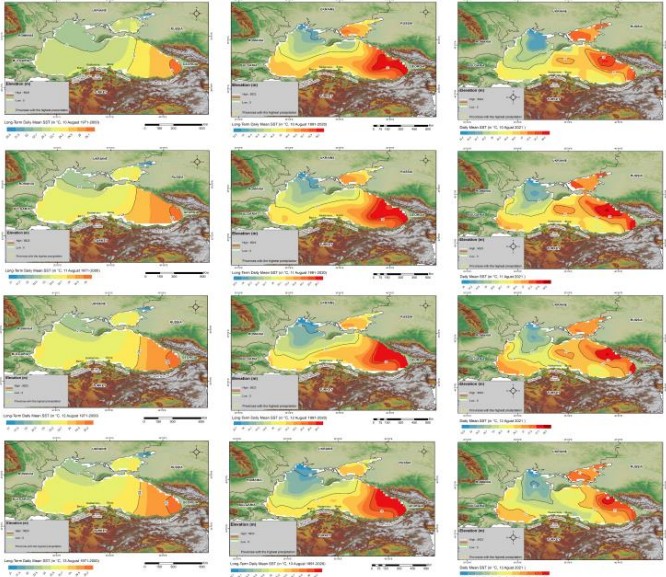






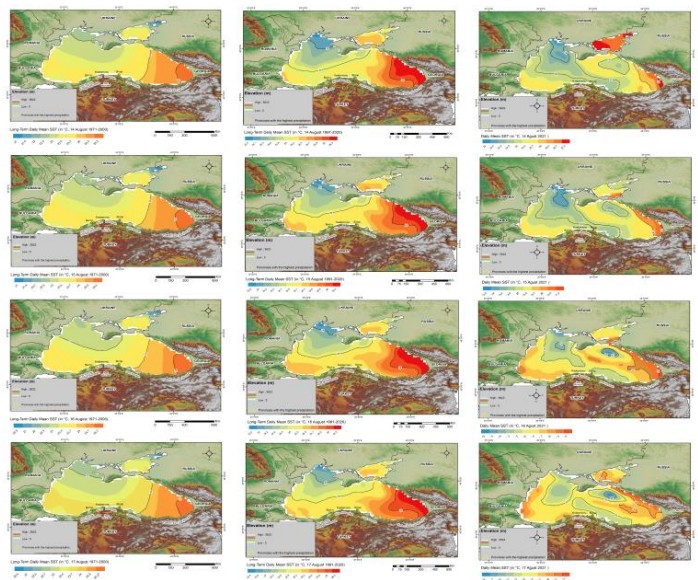


**Figure 6.** The first column is 10-17 August 1971-2000 SST; the Second column is 10-17 August 1991-2020 SST, and the third column is 10-17 August 2021 SST, respectively.

### 4.4. Physiographic Conditions

In often mountainous regions where the precipitation gradient is complex and station distributions are not sufficient, it is crucial to understand the influence of topography in order to be able to interpolate precipitation spatially (Johansson & Chen, 2003). Mountains that cut the general circulation vertically cause orographic precipitation by forcing mobile depressions and front systems to rise, while precipitation tends to decrease depending on the height of the mountain on leeward slopes (Gönençgil, 2009; Smith, 1979). Studies conducted in this context show that the precipitation intensity also increases depending on the speed of forcing the air parcel to rise and the slope of the mountain (Smith, 1979; Weston & Roy, 1994). For example, on November 3, 1987, 817 mm of precipitation was measured in the Gulf of Valencia within 24 hours. In the same region, in the storm in October 1982, 632 mm of precipitation was measured in 24 hours. The common feature of these precipitations is that the moist air coming from the sea is channelled into the valleys and forced to rise. While these extreme precipitations created a flood, the extreme precipitation experienced in October 1982 caused the collapse of the Tous dam (Peñarrocha et al., 2002). Therefore, physiographic conditions are of great importance in extreme precipitation. In order to better evaluate the extreme conditions in the study area, daily precipitation data of a total of 56 stations located in Sinop, Kastamonu and Bartın were used. The number and distribution of stations are sufficient spatial resolution and reflect the orographic effect.




Considering the total precipitation distribution of 45 stations located on the northern and southern slopes of the Küre
Mountains, which exhibit an east-west directional orographic barrier in the study area, it was observed that there were
differences in extreme precipitation values (Fig. 7). In the total precipitation distribution of 45 stations between 10-12
August, it can be seen that the highest precipitation is located on the northern slopes and decreases on the southern
slopes as one move towards the interior (Fig. 7). 3-day precipitation totals to understand the topographical effect were
listed according to the altitude and aspect conditions of the stations. It can be clearly stated that the air masses
channelled into the valleys and forced to orographic uplift quickly create adequate precipitation. For this situation, a
terrain model was made to show the orographic effect at stations 19030 (Devrakani/Kuzköy) and 19032
(Bozkurt/Mamatlar Village) located around the Ezine Stream, where the flood was experienced and where the
maximum rainfall was recorded (Fig 8). The black arrows on the model, representing the winds, are based on the wind
directions of the times when MGM is measuring, and the maximum hourly precipitation is seen (Fig. 7). For example,
the total precipitation values measured in Devrakani/kuzköy and Bozkurt/Mamatlar were three times higher than the
sum of the extreme values measured at the coastal stations.


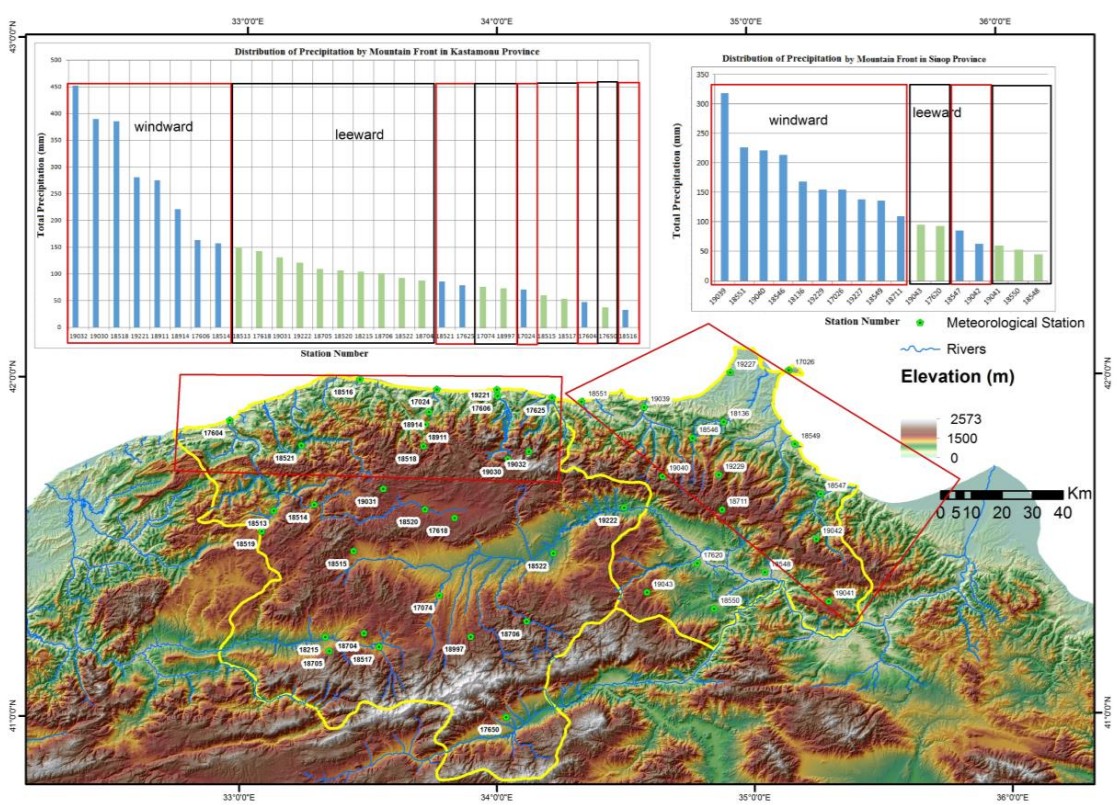

**Figure 7.** Distribution of 45 stations to reflect the orographic effect. The red boxes roughly represent the northern slopes (winward). The stations that are distributed outside the box are the leeward stations behind the orographic barrier. In addition, the total precipitation of these stations on 10-12 August is shown with a column chart.

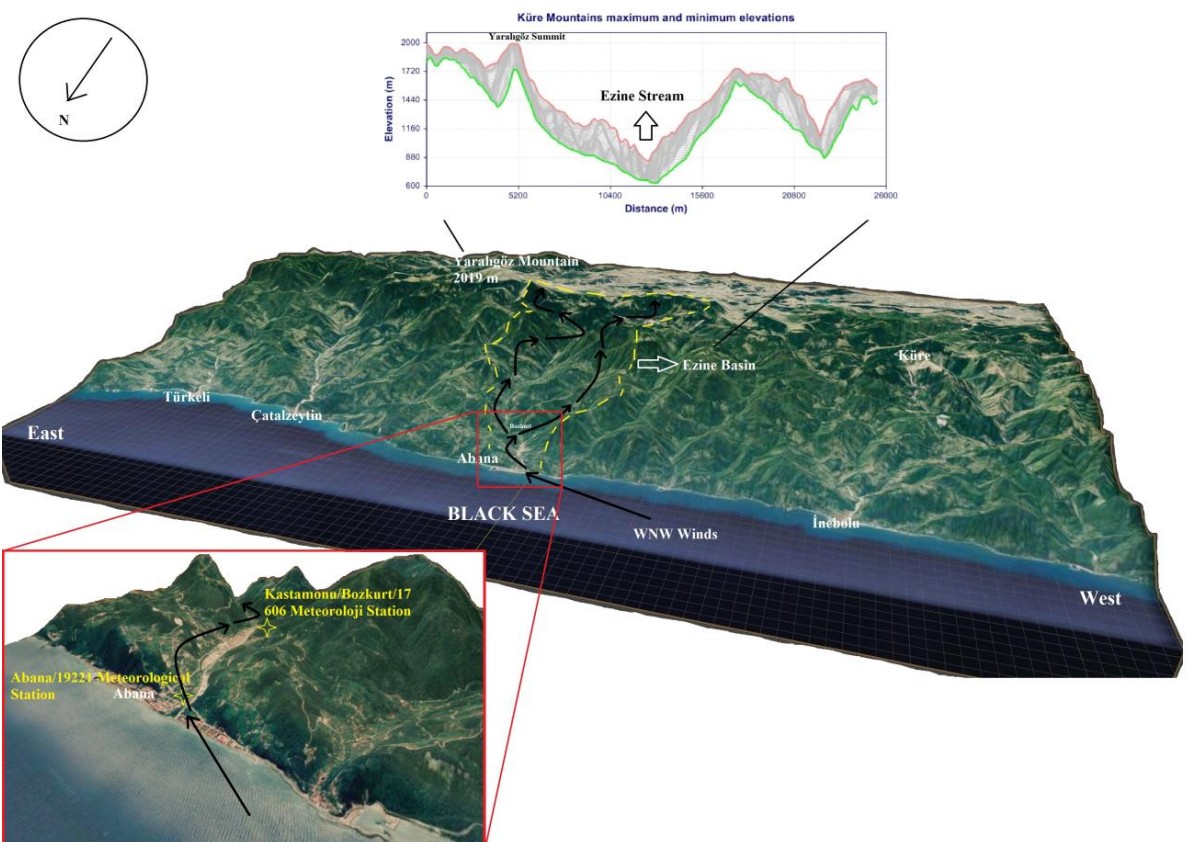

**Figure 8.** Elevation profile (maximum and minimum) of the northern slopes of the Küre Mountains, where the maximum precipitation was measured. 19030 (Devrakani/Kuzköy) and 19032 (Bozkurt/Mamatlar Village) stations are located on the northern slopes of Yaralıgöz Hill at an altitude of 1500 meters. Arrows represent the prevailing wind direction of Abana Meteorology station and Kastamonu Bozkurt station on August 11 (From © Google Maps and SRTM data via BlenderGIS).

On August 11, the cold air coming from the northwest of the Black Sea Basin moved from west to east, following the morphology of the coastline. To the east of Sinop, the air mass turned north and moved towards the cyclone's centre. Wind directions were west, northwest and west-northwest along the coastline in Amasra, Kurucaşile, İnebolu, Abana, Çatalzeytin and Türkeli between 08:00 - 09:00 UTC. Winds in this direction can quickly move from the mouths of the valleys to the interior of the land. The maximum precipitation falling in the region on August 11 was in the coastal line from Bartın to the east of Sinop and the N-S directional valleys. Maximum hourly precipitation measurements observed between 08:00 and 10:00 in Bozkurt/Mamatlar and Devrakani/Kuzköy are closely related to this situation (Fig. 9).


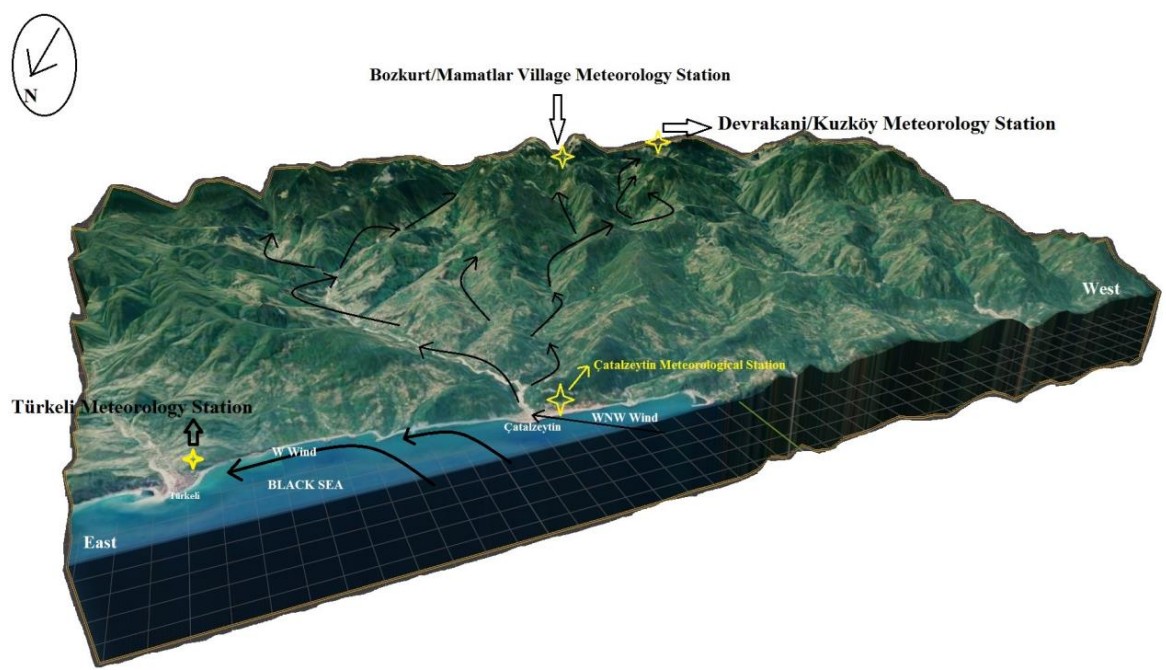


**Figure 9.** Arrows represent west-northwest winds for Türkeli and Çatalzeytin stations between 08:00 and 09:00 UTC on August
11. 61 mm of precipitation was measured in Turkeli on August 11 at 09:00 UTC. Wind data could not be measured in Bozkurt
Mamatlar and Devrekani Kuzköy for these dates, and in terms of channelling into valleys, north for Mamatlar station and
predominantly east and northeast winds for Devrekani Kuzköy must have been effective (From © Google Maps and SRTM data via
BlenderGIS).
**4.5. Extreme Precipitation Analysis**
For the extreme precipitation analysis, hourly precipitation was first evaluated, and then daily evaluations were made.
Precipitation intensity was calculated from hourly data and evaluated according to five rainfall intensity classes
recommended by MGM (Table 2). The highest hourly precipitation in the study area was measured on August 11.
Hourly precipitation values measured between 06:00 and 09:00 UTC on August 11 in Bozkurt/Mamatlar were over
45 mm and are defined as heavy precipitation according to the precipitation intensity class. The precipitation measured
at 10:00 at this station corresponds to extreme precipitation. Devrakani/Kuzköy meteorological station showed similar
values and reached 60 mm, especially at 09:00 UTC. In terms of hourly extremes, 70 mm of precipitation was
measured one hour (on August 10, 22:00 UTC) at Ulus/Ceyüpler station (Fig. 10). This value is very close to the
heavy rainfall threshold value.




**Table 2:** Precipitation intensity classification according to MGM.

| Precipitation Intensity | Hourly Rainfall Amount |
|---|---|
| Light precipitation | 1- 5 mm |
| Moderate Precipitation | 6- 20 mm |
| heavy Precipitation | 21- 50 mm |
| Very Heavy Precipitation | 51- 75 mm |
| violent Precipitation | 76-100 mm |
| Excessive precipitation | Over 100 mm |

In order to determine the precipitation intensity levels suitable for the conditions of Turkey, different precipitation
amounts and severity levels were separated in various climatological studies (Ardel et al., 1969; Çiçek, 2001; Dönmez,
1990; Koçman, 1993). In this study, a different classification was used in order to provide the opportunity to evaluate
all the precipitations of more than a specific value. Thus, with the IDW (Inverse distance weighting) interpolation
method, the daily measured precipitation for 10, 11 and 12 August and then the total precipitation for today were
mapped (Fig 11, 12 and 13). In the Black Sea precipitation regime, daily maximum precipitation values of 431.5 mm
in Zonguldak and 240.9 mm in Rize have been measured (Koçman, 1993). However, on August 11, 2021, 297.4 mm
of daily precipitation was measured for Kastamonu province. As a result of the precipitation that continued for three
days, 452, 318 and 236 mm precipitation were measured in the Kastamonu, Bartın and Sinop provinces, respectively.
These provinces are among the highest measured values to date. (Figures 11, 12 and 13). In addition, the precipitation
in Bartın province was represented with the highest values in this regime as hourly precipitation according to the
MGM records. Flooding occurred in these three provinces due to excessive rainfall. Especially in Kastamonu-Bozkurt,
the settlements in the valley brought great losses (Fig.16).

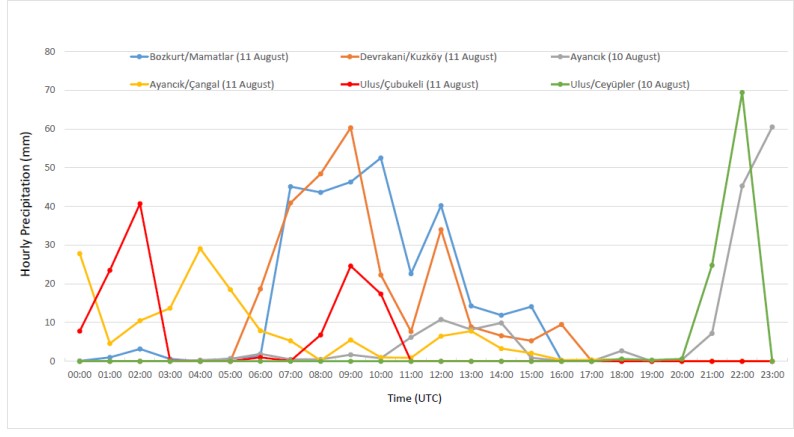





**Figure 10.** Hourly precipitation measurements for August 10-11, 2021.

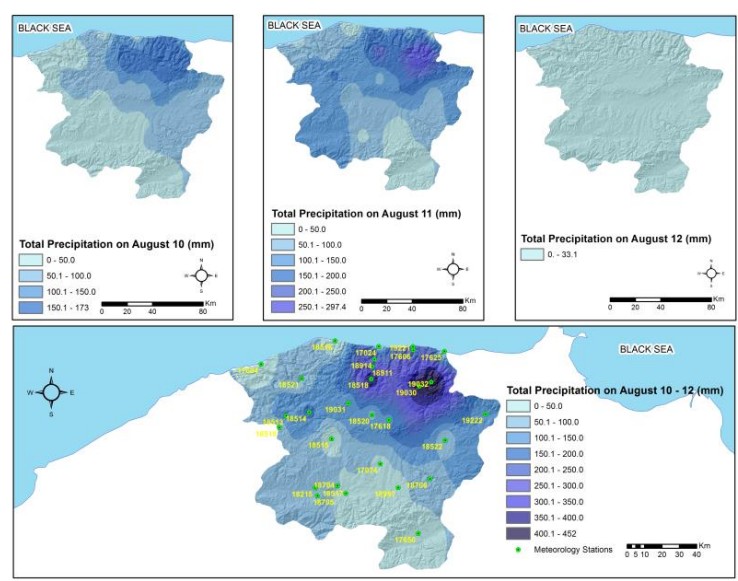


**Figure 11.** The daily precipitation distribution of the province of Kastamonu on 10-12 August.

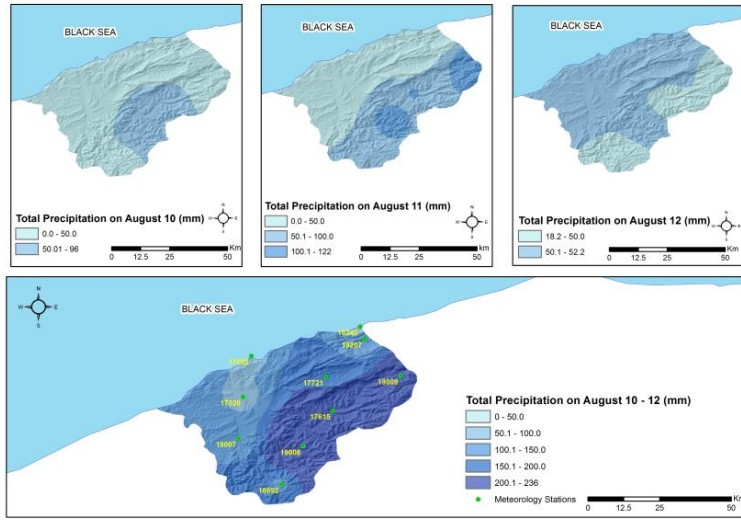


**Figure 12**. The daily precipitation distribution of the province of Bartın on 10-12 August.



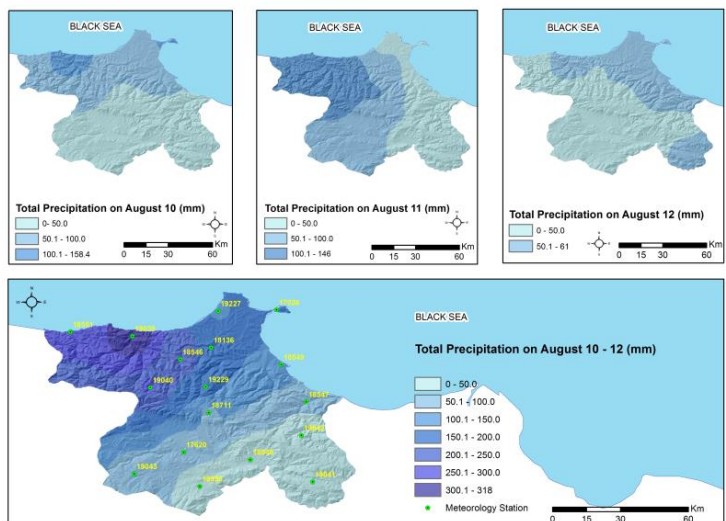

**Figure 13**. The daily precipitation distribution of the province of Bartın on 10-12 August.
**4.5.1. Wind Analysis**
During the dates of extreme precipitation, the dominant wind direction mainly was W-N-NW direction winds to the
atmospheric conditions of that day. Precipitation increased during the hours when the warm and humid northern winds
over the Black Sea were effective. On the contrary, the other important wind direction during the hours of extreme
precipitation was southerly winds. According to the daily temperature data obtained from NOAA, on August 10, 2021,
the minimum temperature was 15.2, the maximum temperature was 28.4, and the average temperature was 25.6 °C at
Kastamonu Meteorology Station. Especially after noon, the temperature dropped to 17°C. On August 11, 2021, the
maximum temperature was measured at 18 degrees. These values are also supported by the SEVIRE MSG data with
synoptic observation (Figures 14a, 14b and 14c). The temperature on the coasts of Sinop was 24 °C. These temperature
values reached lower on August 12, 2021 (Figure 14c). During these dates, the temperature difference between the
land and the sea approached 10 °C. As a result, a clear contrast has occurred in the meteorological elements in a short
period, which is expressed as discontinuity. This situation must have triggered the southerly winds. This must have
affected the relatively warm and humid air coming from the north and the colder and drier air to meet each other
mainly channelled into the valleys. As a result, extreme precipitation was recorded during the hour intervals when the
southerly winds blew (Figure 15). Previous studies have determined that this mechanism effectively causes excessive
precipitation on the Eastern Black Sea coast (Baltaci, 2017).



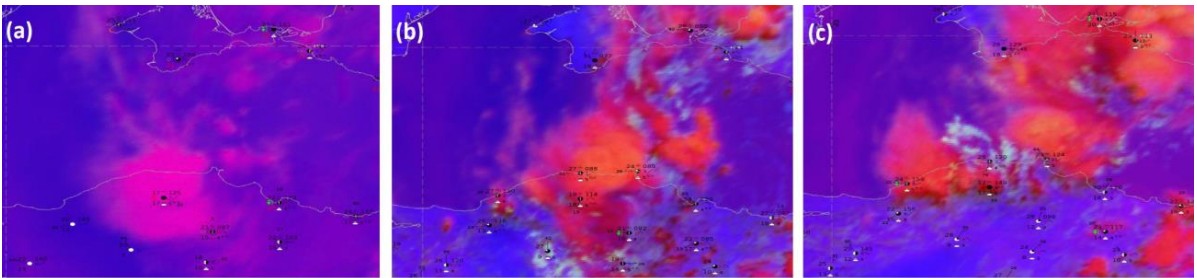


**Figure 14.** (a) Convection RGB product from SEVIRE MSG (Meteosat Second Generation) together with SYNOP observations, 2021-08-10 18:00Z. (b) 2021-08-10 12:00Z. (c) 2021-08-12 12:00Z. Source: EUMETSAT (https://www.eumetsat.int/).

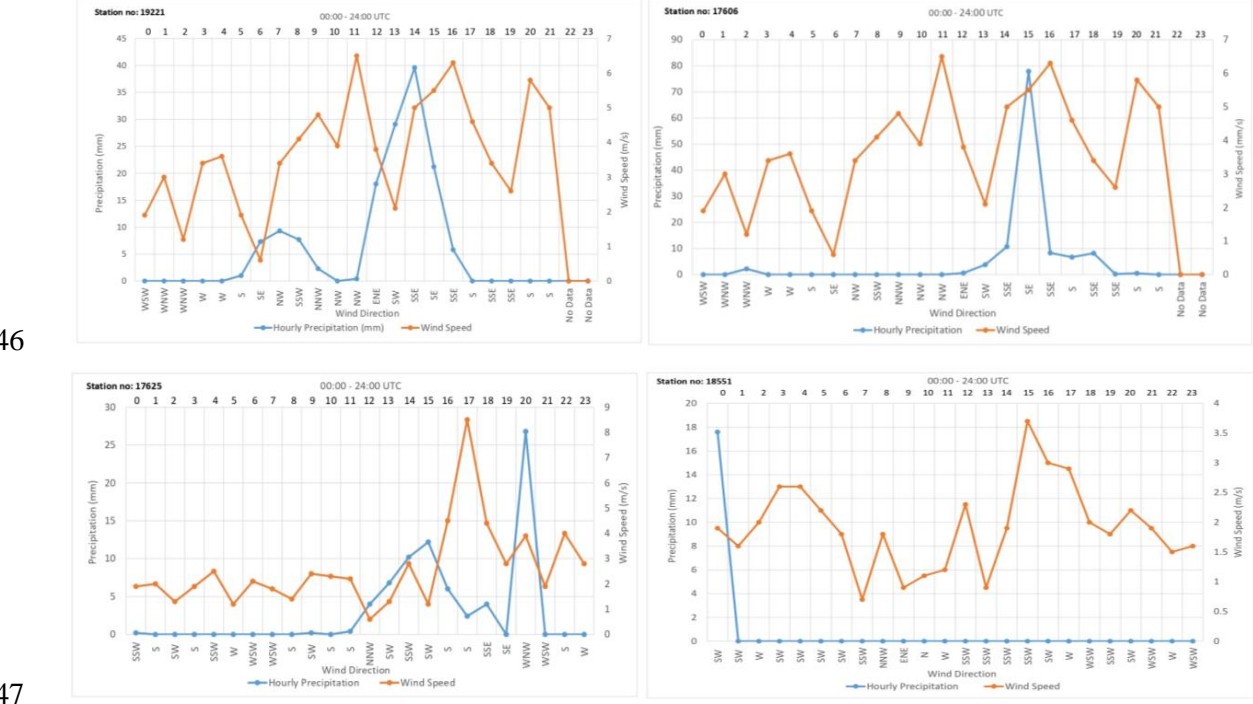







**Figure 15**. Hourly south winds and hourly precipitation values with station numbers. (a) Winds from the south for station 17024
and southeasterly for station 18914. (b) Southeasterly winds for both stations 17606 and 19221. (c) For station 18551, the winds




blew in line with the southwest-trending valley. (d) Southwest and southeast oriented valleys for station 17625 (From © Google Maps
and SRTM data via BlenderGIS).

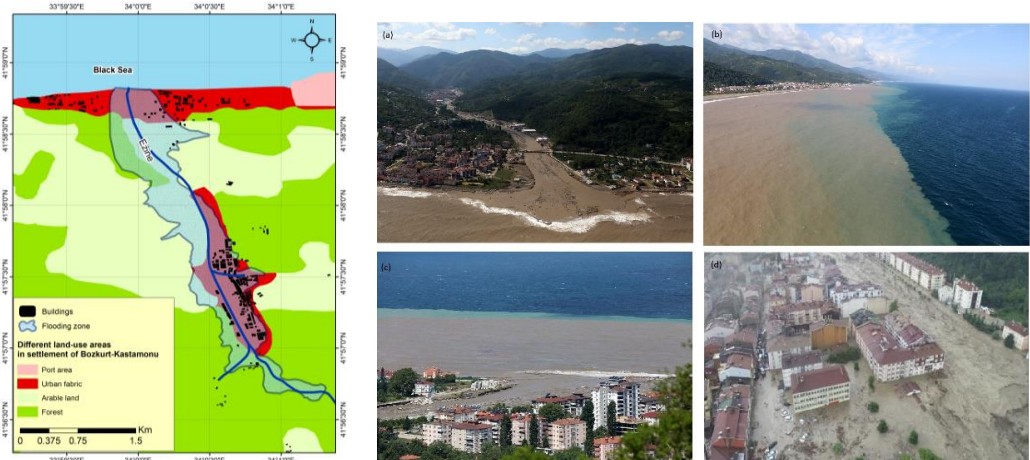

**Figure 16.** Different land use and buildings within the floodplain. Ezine Stream, Bozkurt. Photos showing the destructive effects
of the 11 August 2021 flash floods: (a) The mouth of the Ezine Valley and the entrance of the suspended loads to the Black Sea,
(b and c) progress of suspended loads in the Black Sea, (d) floodplain. Represents the largest flood in the three provinces,
experienced August 10-12, 2021 (https://www.afad.gov.tr/bartin-kastamonu-ve-sinopta-). Photos source: www.ensonhaber.com.
**5 Conclusion**
In this study, the synoptic conditions of extreme precipitation in the provinces of Sinop, Kastamonu and Bartın were
investigated. As a result of the heavy rains that fell on these provinces in three days, Bartın Ulus; Kastamonu Azdavay,
İnebolu, Bozkurt, Küre and Pınarbaşı districts and Sinop Ayancık districts were affected by the flood. According to
the official figures included in the Turkey Ministry of Interior Disaster and Emergency Management Presidency
(AFAD) on 01.09.2021, 82 people (71 in Kastamonu, 10 in Sinop, and 1 in Bartın) lost their lives due to the flood. In
addition, energy, transportation, access to drinking water, communication and shelter areas were damaged by the
flood.
When the synoptic conditions were examined, the high-pressure area moved eastward over Europe on August 10 and
was located northwest and northeast of the Black Sea. The blocking presence of the warm anticyclone over eastern
Europe generated north-northeast winds. The distinguishing feature of this synoptic process is that it stopped its
cyclone over the Black Sea. Because the blocking feature of the static high pressure in the area blocked the roads of
the cyclone to the north and northeast. Since August 10, the pressure gradient of the surrounding areas has increased


with the low pressure developing in the middle parts of the Black Sea (the line between Crimea and Kastamonu). The
activation of the fronts on 10-12 August 2021, the intensification of cyclogenesis on the Black Sea until about 40°
latitude, and the presence of active front formation are also noteworthy. In areas where this low pressure was located
on the Black Sea, the effectiveness of the cyclone increased, with the SST being 27 °C. Thus, the Black Sea gained a
strong low-level convergence feature. In addition to all these, the temperature difference between the sea surface and
850 hPa reached 13 °C. The cold air, which entered the Black Sea from the north-northwest, warmed while moving
towards the low-pressure centre on the Black Sea, moved along the northern coast of Turkey, oriented towards the
Kerch Strait and gained a cyclonic characteristic. As a result of this action, the heated air mass also encountered the
cold air coming over the Turkish lands, increasing the instability conditions and heavy rains were also influential. In
particular, topographic conditions affected the spatial distribution of excessive precipitation.
It is an atmospheric situation that can be expected to experience more such events triggered by the differentiation in
sea-land surface temperatures, with the temperature increases predicted in the coming years. In this context, heavy
rains are expected to increase in the coming years. Applicable plans should be made to protect residential areas from
damage caused by heavy rainfall events. Preventing construction around the floodplains of rivers, especially in areas
where floods and flood disasters have occurred in the past, and revising the zoning plans according to the changing
climatic conditions are among the most critical requirements. In addition, to create a sustainable, healthy, smart and
resilient local society, the residents have to be educated about the risk of disasters, climate adaption and action.
*Author contribution.* O.H prepared the article with contributions from all co-authors. B.G developed the idea for the
article. The B.G and Z.A checked the content of the article and gave necessary directions (methodology, reviewing,
editing of the manuscript). In addition, Z.A produced the 700 hPa wind direction map. All remaining figures and maps
were produced by O.H.
*Data availability.* The following data sets are publicly available. Synoptic charts of the daily event are available at
https://mgm.gov.tr/sondurum/guncel-haritalar.aspx.    Sea    surface    temperature    data    are    available    at
https://psl.noaa.gov/data/gridded/data.noaa.oisst.v2.highres.html. The satellite data used for the case study are
available at http://www.eumetrain.org/ and https://view.eumetsat.int/productviewer?v=default. Additionally, daily
temperature data is available at https://www.ncei.noaa.gov/maps/daily-summaries/ and August 11, 2021, 700 hPa u
and v direction vector wind data available at https://psl.noaa.gov/data/gridded/data.ncep.reanalysis.html. Land use
data are available at https://land.copernicus.eu/pan-european/corine-land-cover/clc2018?tab=download. You can
contact the responsible author about other data you are interested in.
*Competing interests.* The authors declare that there is no conflict of interest.





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
