# Peer review of "An atmospheric approach to the flood disaster in the Western Black Sea region (Turkey) on 10-12 August 2021"

_Natural Hazards and Earth System Sciences, 2022_

## Referee Comment (RC1)

Manuscript: https://doi.org/10.5194/nhess-2022-185

**Title:** An atmospheric approach to the flood disaster in the Western Black Sea region (Turkey) on 10-12 August 2021

**Summary and Comments to the Authors**

As far as I can tell (see below), this paper aims to analyse "An atmospheric approach to the flood disaster in the Western Black Sea region (Turkey) on 10-12 August 2021" with the aid of precipitation measurements, weather maps and satellite images.

I said "as far as I can tell". Because, the paper is very poorly written: as a result, it is difficult to understand what the authors are trying to convey or how they've done it. The problem is partly due to poor English: I realise that the authors are not native English speakers, but there are many places where I honestly have no idea what they are trying to say. This is often due to the use of vocabulary which is meaningless in English, but there are also many grammatical and linguistic errors: as a result, the paper is very hard to read. Authors also do not use atmospheric science terminology properly in some places, which makes it harder to follow.

However, the problems are not just with the English. I also think (i) that the work in the paper is limited; (ii) that the results are reported poorly; and (iii) that they are not linked well to the stated aims of the paper. Some specific comments are given below:

1. The results are based on the analysis of the pre-ready weather maps and satellite images, and some statements are made but no supporting numerical studies about them. Authors make some statements but there is no numerical to study to support them. For example; in 27-29 authors state that "Thus, the developing convective cell gradually became more vigorous, expanded its area of influence in the interior of the land due to the cyclone movement as of the morning of August 10 and caused heavy rains with the effect of the topography." But this was not shown in related section, just some statements were made.

2. Lines 19-20; "The Basra low pressure settled over the eastern Black Sea during this period." What is "the Basra low", is there a published climatological study about it?

3. Lines 24-29; "On August 10 2021, the surface temperature of the Black Sea, especially the eastern part, reached 28.3°C (3.5°C higher than average)." How is the average is defined, climatological average, if yes, which period?

4. Lines 116-117; "When the synoptic situation at ground level is examined on August 9, it is seen that the Basra low pressure extends to the Eastern Black Sea Region." It seems to be an over statement, it looks like an extended ridge.

5. Lines 123-124; "The blocking presence of the warm anticyclone over eastern Europe generated northerly winds (Fig. 4)." Not every anticyclone is a blocking high, what makes authors to make this statement?

6. Line 124; "The distinguishing feature of this synoptic process is that it stopped the cyclone over the Black Sea." It is not clear what causes this? The high on the west or on the east?

7. Lines 125-126; authors state a mechanism but do not explain anything and refer to a paper. It is not readers job to read referenced papers to figure out what authors mean.

8. Lines 128-132; authors make some statements but do not explain by referring to existing figures or provide a figure that support their statements.

9. Lines 133-134; "In addition to all these, the temperature difference between sea surface temperatures and 850 hPa is around 13 °C (Fig. 3e)." It is not clear on figure- 3e and not clearly stated in the figure caption what Fig3e displays?

10. Lines 134-135; It is not clear on the figure and not clearly stated in the figure caption what Fig3f displays? There seems to be no isotherms plotted, but authors refer to temperature gradients.

11. Lines 135-136; "When the relatively cold northerly winds entered the Black Sea, they warmed up from the bottom and the moisture content increased, resulting in an instability condition." Authors do not provide any upper level sounding charts to support this statement, they only make conceptual description of plausible mechanisms.

12. Lines 143-146; Figure 3 caption is not clear, poorly written

13. Lines 158-161; "This is due to the clouds rising to the troposphere border, especially when the lower layers of the air mass are hot and strong vertical air movement develops. Cold and thick convective clouds with large ice particles on top, e.g. Cb tops, appear red (Figure 5d, 5e, 5f). This is important for detecting rising movements and more severe weather conditions." Authors also need to provide upper air soundings to support this statement.

14. Lines 164-167; "Unlike Day Microphysics RGB, convection zones can be observed more easily in Day Convective Storms RGB. Severe convective storms appear bright yellow in this colour scheme (Figure 5g, 5h, 5i). High-level thick ice clouds and convective CB clouds containing large ice crystals appear red, and convective clouds containing tiny ice crystals appear yellow." Need to be explained clearly, authors only refer to figures and expect readers to evaluate each of them. What is CB, is it explained until this section? Also, no reference was given for colours and related cloud formation.

15. Lines 170-174; This part need to be explained better by referring each figure and relating to cloud developments, related vertical and horizontal motions.

16. Lines 194-195;" In our study, where the temperature difference between the sea surface and the upper atmosphere is excellent, many studies emphasize the effect of sea or lakes on precipitation where these temperature differences are emphasized (Suriano & Leathers, 2017)." Which upper atmosphere, which level?

17. Lines 199-200; "Especially in cases where the temperature difference between the surface temperature and 850 hPa is 13 °C and above, the lake effect has been accepted as the threshold value for the development of precipitation (Holroyd III, 1971; Kunkel et al., 2002; Niziol, 1987)." The temperature threshold (13 °C) between the sea (or lake) surface and the 850 hPa level is used for "sea-effect snow" and the given references refer to lake-effect storm cases. This threshold is usually used for sea-effect snow type of precipitation, is it also relevant for rain, which is a warm type of precipitation compared to snow?

18. Lines 216-218; "According to MGM data, this situation dynamically created a breeze of 3 m/s from land to sea due to the increase in the temperature difference between the sea and the land during the night and morning hours." Is this 10-meter wind or which level of the atmosphere?

19. Lines 218-220; "This triggered a low level of horizontal convergence and allowed the air to rise. Thus, the divergence at the upper level contributes to moving the low-level horizontal flow and humidity to the higher level." Authors make statements on convergence and air parcel's upward motion but do not show supporting figures. Also, authors should explain this by referring to related atmospheric levels and related figures.

20. Line 121; Authors do not show any information for supporting for this statement, how instability is triggered?

21. Line 122; "Between 10-17 August 1971-2000, the maximum temperatures were 25 °C," Are these averaged 2-meter maximum temperatures? Are they based on the stations listed in Table 1? Or are they SST distributions depicted in Figure 6?

22. Line 228; "Figure 6 is not clear; it is hard to follow authors' findings on the figure. Also, there are two main figures and they are not well explained by figure captions.

The manuscript is hard to follow and it is hard to report in here every line of the text, therefore I shall not continue further.

**General comments and decision**

The authors do not engage properly with literature or explained it well. As a result of these concerns, I cannot agree with the authors' claims (lines 8-29).

I have several other concerns about the paper, although they are not all listed in here. However, my comments above should make it clear already that I do not think it is appropriate for publication in any reputable journal. I am sorry that I cannot be more positive on this manuscript.